# Variation in antibiotic prescription rates in febrile children presenting to emergency departments across Europe (MOFICHE): A multicentre observational study

Nienke N. Hagedoorn[1], Dorine M. Borensztajn[1], Ruud Nijman[2], Anda Balode[3], Ulrich von Both[4,5], Enitan D. Carrol[6,7], Irini Eleftheriou[8], Marieke Emonts[9,10,11], Michiel van der Flier[12,13,14], Ronald de Groot[12,13], Jethro Herberg[2], Benno Kohlmaier[15], Emma Lim[9,10], Ian Maconochie[16], Federico Martinon-Torres[17], Daan Nieboer[18], Marko Pokorn[19], Franc Strle[19], Maria Tsolia[8], Shunmay Yeung[20], Dace Zavadska[3], Werner Zenz[15], Clementien Vermont[21], Michael Levin[2], Henriëtte A. Moll[1]*, on behalf of the PERFORM consortium[¶]

1 Department of General Paediatrics, Erasmus MC–Sophia Children's Hospital, Rotterdam, the Netherlands, 2 Section of Paediatric Infectious Disease, Imperial College London, London, United Kingdom, 3 Department of Paediatrics, Children's Clinical University Hospital, Rīgas Stradiņa Universitāte, Riga, Latvia, 4 Division of Paediatric Infectious Diseases, Dr. von Hauner Children's Hospital, University Hospital, Ludwig Maximilian University, Munich, Germany, 5 Partner Site Munich, German Center for Infection Research (DZIF), Munich, Germany, 6 Institute of Infection and Global Health, University of Liverpool, Liverpool, United Kingdom, 7 Alder Hey Children's NHS Foundation Trust, Liverpool, United Kingdom, 8 Second Department of Paediatrics, P. & A. Kyriakou Children's Hospital, National and Kapodistrian University of Athens, Athens, Greece, 9 Paediatric Immunology, Infectious Diseases & Allergy, Great North Children's Hospital, Newcastle upon Tyne Hospitals NHS Foundation Trust, Newcastle upon Tyne, United Kingdom, 10 NIHR Newcastle Biomedical Research Centre, Newcastle upon Tyne Hospitals NHS Trust and Newcastle University, Newcastle upon Tyne, United Kingdom, 11 Translational and Clinical Research Institute, Newcastle University, Newcastle upon Tyne, United Kingdom, 12 Paediatric Infectious Diseases and Immunology, Amalia Children's Hospital, Radboud University Medical Center, Nijmegen, the Netherlands, 13 Section of Paediatric Infectious Diseases, Laboratory of Medical Immunology, Radboud Institute for Molecular Life Sciences, Nijmegen, the Netherlands, 14 Paediatric Infectious Diseases and Immunology, Wilhelmina Children's Hospital, University Medical Center Utrecht, Utrecht, the Netherlands, 15 Department of General Paediatrics, Medical University of Graz, Graz, Austria, 16 Paediatric Emergency Medicine, Imperial College Healthcare NHS Trust, London, United Kingdom, 17 Genetics, Vaccines, Infections and Paediatrics Research Group (GENVIP), Hospital Clínico Universitario de Santiago de Compostela, Santiago de Compostela, Spain, 18 Department of Public Health, Erasmus University Medical Center, Rotterdam, the Netherlands, 19 Department of Infectious Diseases, University Medical Centre Ljubljana, Ljubljana, Slovenia, 20 Faculty of Infectious and Tropical Diseases, London School of Hygiene & Tropical Medicine, London, United Kingdom, 21 Department of Paediatric Infectious Diseases and Immunology, Erasmus MC–Sophia Children's Hospital, Rotterdam, the Netherlands

¶ Membership of the PERFORM consortium is provided in the Acknowledgements.
* h.a.moll@erasmusmc.nl

**Data Availability Statement:** An anonymized data set containing individual participant data is

## Abstract

### Background

The prescription rate of antibiotics is high for febrile children visiting the emergency department (ED), contributing to antimicrobial resistance. Large studies at European EDs covering diversity in antibiotic and broad-spectrum prescriptions in all febrile children are lacking. A better understanding of variability in antibiotic prescriptions in EDs and its relation with viral or

available in a public data repository: https://data.hpc.imperial.ac.uk/resolve/?doi=7251. DOI: 10.14469/hpc/7251. For inquiries to obtain the full dataset, please contact the data manager of the PERFORM consortium (Tisham.de08@imperial.ac.uk).

**Funding:** This project has received funding from the European Union's Horizon 2020 research and innovation programme under grant agreement No. 668303. The Research was supported by the National Institute for Health Research Biomedical Research Centres at Imperial College London, Newcastle Hospitals NHS Foundation Trust and Newcastle University. The views expressed are those of the author(s) and not necessarily those of the NHS, the NIHR or the Department of Health. For the remaining authors no sources of funding were declared. The funders had no role in study design, data collection and analysis, decision to publish, or preparation of the manuscript.

**Competing interests:** I have read the journal's policy and the authors of this manuscript have the following competing interests: MvdF received a grant from CSL Behring preclinical work immunoglobulin formulations; FS served on the scientific advisory board for Roche on Lyme disease serological diagnostics, received research support from the Slovenian Research Agency [grant number P3-0296], and is an unpaid member of the steering committee of the ESCMID Study Group on Lyme Borreliosis/ESGBOR. The other authors have declared that no competing interests exist.

**Abbreviations:** CRP, C-reactive protein; ED, emergency department; MOR, median odds ratio; RTI, respiratory tract infection.

bacterial disease is essential for the development and implementation of interventions to optimise antibiotic use. As part of the PERFORM (Personalised Risk assessment in Febrile illness to Optimise Real-life Management across the European Union) project, the MOFICHE (Management and Outcome of Fever in Children in Europe) study aims to investigate variation and appropriateness of antibiotic prescription in febrile children visiting EDs in Europe.

## Methods and findings

Between January 2017 and April 2018, data were prospectively collected on febrile children aged 0–18 years presenting to 12 EDs in 8 European countries (Austria, Germany, Greece, Latvia, the Netherlands [$n = 3$], Spain, Slovenia, United Kingdom [$n = 3$]). These EDs were based in university hospitals ($n = 9$) or large teaching hospitals ($n = 3$). Main outcomes were (1) antibiotic prescription rate; (2) the proportion of antibiotics that were broad-spectrum antibiotics; (3) the proportion of antibiotics of appropriate indication (presumed bacterial), inappropriate indication (presumed viral), or inconclusive indication (unknown bacterial/viral or other); (4) the proportion of oral antibiotics of inappropriate duration; and (5) the proportion of antibiotics that were guideline-concordant in uncomplicated urinary and upper and lower respiratory tract infections (RTIs). We determined variation of antibiotic prescription and broad-spectrum prescription by calculating standardised prescription rates using multilevel logistic regression and adjusted for general characteristics (e.g., age, sex, comorbidity, referral), disease severity (e.g., triage level, fever duration, presence of alarming signs), use and result of diagnostics, and focus and cause of infection. In this analysis of 35,650 children (median age 2.8 years, 55% male), overall antibiotic prescription rate was 31.9% (range across EDs: 22.4%–41.6%), and among those prescriptions, the broad-spectrum antibiotic prescription rate was 52.1% (range across EDs: 33.0%–90.3%). After standardisation, differences in antibiotic prescriptions ranged from 0.8 to 1.4, and the ratio between broad-spectrum and narrow-spectrum prescriptions ranged from 0.7 to 1.8 across EDs. Standardised antibiotic prescription rates varied for presumed bacterial infections (0.9 to 1.1), presumed viral infections (0.1 to 3.3), and infections of unknown cause (0.1 to 1.8). In all febrile children, antibiotic prescriptions were appropriate in 65.0% of prescriptions, inappropriate in 12.5% (range across EDs: 0.6%–29.3%), and inconclusive in 22.5% (range across EDs: 0.4%–60.8%). Prescriptions were of inappropriate duration in 20% of oral prescriptions (range across EDs: 4.4%–59.0%). Oral prescriptions were not concordant with the local guideline in 22.3% (range across EDs: 11.8%–47.3%) of prescriptions in uncomplicated RTIs and in 45.1% (range across EDs: 11.1%–100%) of prescriptions in uncomplicated urinary tract infections. A limitation of our study is that the included EDs are not representative of all febrile children attending EDs in that country.

## Conclusions

In this study, we observed wide variation between European EDs in prescriptions of antibiotics and broad-spectrum antibiotics in febrile children. Overall, one-third of prescriptions were inappropriate or inconclusive, with marked variation between EDs. Until better diagnostics are available to accurately differentiate between bacterial and viral aetiologies, implementation of antimicrobial stewardship guidelines across Europe is necessary to limit antimicrobial resistance.

## Author summary

### Why was this study done?

- Respiratory infections, which are mainly caused by viruses, account for the majority of antibiotic use in children. In children with respiratory infections, antibiotic prescription rates vary across emergency departments (EDs) in Europe.

- In order to optimise antibiotic prescriptions, it is important to better understand variability and appropriateness in antibiotic prescriptions.

### What did the researchers do and find?

- In this prospective observational study, we included routine information of 35,650 children (median age 2.8 years) with fever attending 12 different EDs in Europe and calculated the proportion of antibiotic prescriptions and broad-spectrum antibiotic prescriptions. We adjusted for differences in population including age, comorbidity, disease severity, and focus and cause of infection.

- Across EDs, antibiotic prescription rates ranged between 22.4% and 41.6%, and of these prescriptions, broad-spectrum antibiotic rates ranged between 33.0% and 90.3%. Standardised antibiotic prescription rates ranged between 0.77 and 1.35, and standardised rates of broad-spectrum antibiotics ranged between 0.65 and 1.75.

- Prescriptions that were inappropriately indicated ranged from 0.6% to 29.3%, and inconclusive prescriptions ranged from 0.5% to 61.7%. The proportion of oral prescriptions with inappropriate duration ranged from 4.4% to 59.0%.

### What do these findings mean?

- In this study we found variation of prescription of antibiotics and broad-spectrum antibiotics between EDs in children with fever, even when correcting for age, comorbidity, disease severity, diagnostics, and focus and cause of infection.

- Variation was especially large in prescriptions for viral infections and infections of unknown cause.

- In this cohort of febrile children, one-third of prescriptions were of inappropriate or inconclusive indication, with variation between EDs. In addition, guideline concordance for respiratory and urinary infections varied widely across EDs.

- Generalisation of these results to all EDs in Europe should be undertaken with caution.

- Implementation of guidelines is needed to improve appropriate prescription of antibiotics, whilst new biomarkers will further improve antibiotic prescription.

## Introduction

Fever is one of the most common reasons for children to visit the emergency department (ED), and most visits are accounted for by self-limiting infections [1,2]. The proportion of

children with a serious bacterial infection that needs treatment with antibiotics ranges from 7% to 13%, while antibiotic prescription rates in febrile children at EDs are between 19% and 64% [3–5]. Inappropriate antibiotic use, including the unnecessary use of broad-spectrum antibiotics, remains high in children, promoting the emergence of antimicrobial resistance [6–9]. Inappropriate antibiotic prescriptions were described in around 30% of outpatient prescriptions. However, these outpatient settings mainly involve primary care, and limited studies are available on specific emergency care [6,10].

Large variability exists between countries in antibiotic prescriptions in inpatient and outpatient settings, according to several large studies [6,8,11–14]. In general, these large studies did not adjust for differences in populations. In children, previous studies have demonstrated substantial variation of antibiotic use in general outpatient settings in the United States and Europe, indicating possible overuse of antibiotics [5,6,10,15].

A literature review on antibiotic prescription rates and their determinants in febrile children in emergency care found large heterogeneity of studied populations, which limited the ability to draw conclusions [16]. One recent European study, focusing solely on EDs, showed significant differences in antibiotic prescription rates in otherwise healthy children with respiratory tract infections (RTIs) [5]. Large studies at EDs across Europe are lacking that cover antibiotic and broad-spectrum prescriptions in all febrile children, including patients with comorbidity, patients with detailed clinical information, and patients in different diagnostic groups. Additionally, previous studies have addressed appropriate prescribing based on diagnosis coded with the International Classification of Diseases [6,10,17]. This classification, however, may not accurately take into account bacterial versus viral aetiology. Antibiotic prescription rates for viral and bacterial disease using a structured classification have not yet been investigated at EDs.

A better understanding of variability in antibiotic prescriptions in EDs and its relation with bacterial or viral disease, taking into account differences in case mix, is essential for the development and implementation of interventions to optimise antibiotic use. In addition, knowledge regarding variation of prescribing in infections where antibiotic prescription is inappropriate, such as prescriptions in viral disease, prescriptions of inappropriate duration, or prescriptions that are not concordant with guidelines, could target and improve implementation of antimicrobial stewardship guidelines at the ED level.

In this study, we aim to investigate the variation and appropriateness of rates and types of antibiotic prescription in febrile children attending 12 different EDs in Europe.

This is a main analysis of the MOFICHE (Management and Outcome of Fever in Children in Europe) study, which is embedded in the PERFORM (Personalised Risk assessment in Febrile illness to Optimise Real-life Management across the European Union) project (https://www.perform2020.org) [18]. MOFICHE is an observational multicentre study that studies the management and outcome of febrile children in Europe using routine data. The overall aim of PERFORM is to improve management of febrile children and to improve diagnosis through development of new diagnostic tests to discriminate viral and bacterial infections in children.

## Methods

### Study design

MOFICHE is a prospective observational study using data that are collected as part of routine care. This study is reported as per the Strengthening the Reporting of Observational Studies in Epidemiology (STROBE) guideline (S1 Text), and data were analysed using an a priori statistical analysis plan (S2 Text). The study was approved by the ethics committees in the participating hospitals, and the need for informed consent was waived (S3 Text).

## Study population and setting

Children aged 0–18 years presenting with fever (temperature $\geq$ 38.0˚C) or a history of fever (fever within 72 hours before ED visit) were included. Twelve EDs from 8 European countries participated in this study: Austria, Germany, Greece, Latvia, the Netherlands ($n$ = 3), Spain, Slovenia, and the United Kingdom ($n$ = 3). The EDs were included because they all participated in the PERFORM project. Characteristics of these EDs are described in S4 Text and in a previous publication [19]. In short, the participating hospitals were either university hospitals ($n$ = 9) or large teaching hospitals ($n$ = 3), and 11 EDs had paediatric intensive care facilities. Nine EDs were paediatric focused, and 3 EDs served both children and adults. Care for febrile children was supervised by general paediatricians (7 EDs), by paediatric emergency physicians (2 EDs), or by a general paediatrician or a (paediatric) emergency physician (3 EDs). All data were available in electronic healthcare records in 5 EDs, 1 ED used paper records, and 6 EDs used a combination of paper and electronic healthcare records.

Data were collected from January 2017 until April 2018, and for at least 1 year at each site to include all seasons. The period of data collection per month ranged from 1 week per month to the whole month in the participating hospitals (S4 Text).

## Sample size

We expected to include 40,000 children with at most 5% missing data. Pilot data showed an overall antibiotic prescription rate of 30%. Applying 10 events per variable, this study is large enough to analyse over 1,000 determinants for the outcome antibiotic prescription [20]. We performed a post hoc sample size estimation for a desired width of the 95% confidence interval (CI) of standardised antibiotic prescription rate per ED. The expected width of the CI of the standardised prescription rate was below 0.5 for the smallest ED.

## Data collection

Data were collected as part of routine ED care. The local research team entered data from patient records in an electronic case record form (eCRF) [21]. Collected data included age, sex, season, referral, comorbidity (chronic condition expected to last at least 1 year) [22], triage urgency, fever duration, fever measured at ED, presence of "red traffic light" symptoms for identifying risk of serious illness (alarming signs) (from the National Institute for Health and Care Excellence [NICE] guideline on fever [23]: decreased consciousness, ill appearance, work of breathing, meningeal signs, focal neurology, non-blanching rash, dehydration, status epilepticus), previous antibiotic use, vital signs (heart rate, respiratory rate, oxygen saturation, temperature, capillary refill time), laboratory results (white blood cell count, C-reactive protein [CRP], urinalysis), imaging (chest X-ray and other imaging), microbiological investigations (cultures and respiratory viral tests), and disposition (intensive care unit admission, general ward admission or discharge). We collected data on antibiotics prescribed in the ED or started on the first day of hospital admission (type, route of administration, and duration). The focus of infection was categorised as upper respiratory tract (otitis media, tonsillitis/pharyngitis, other), lower respiratory tract, gastrointestinal tract, urinary tract, skin, musculoskeletal, sepsis, central nervous system, flu-like illness, childhood exanthem, inflammatory syndrome, undifferentiated fever, or other.

To date, no reference standard exists to classify the cause of infection in routine ED practice [24]. The PERFORM consortium adapted the consensus-based flowchart from Herberg and colleagues [25,26], combining all available clinical data, investigation results such as CRP, cultures, and imaging. This flowchart was used to define the presumed cause of infection for each patient visit: definite bacterial, probable bacterial, bacterial syndrome, unknown bacterial/

viral, viral syndrome, probable viral, definite viral, trivial, inflammatory syndrome, and other (Fig 1). The diagnosis definite bacterial infection was assigned only when a sterile site culture identified pathogenic bacteria. The diagnosis 'probable bacterial infection' was assigned when a bacterial syndrome was suspected but no bacteria were identified and CRP was above 60 mg/l. Patients with clinical bacterial symptoms and CRP ≤ 60 mg/l or no CRP were classified as 'bacterial syndrome'. Children with suspected viral infections were classified as 'viral syndrome' (no CRP or CRP > 60 mg/l) or 'definite viral' (CRP ≤ 60 mg/l) when a virus was identified that matched the clinical symptoms. Children with a viral syndrome and CRP ≤ 60 mg/l, but no identified virus, were classified as 'probable viral'. Children who did not fit these definitions were classified as unknown bacterial/viral. Children with mixed infections (bacterial and viral co-infection) were classified as bacterial. Children with trivial infections, inflammatory syndrome, or other infections were classified as 'other'.

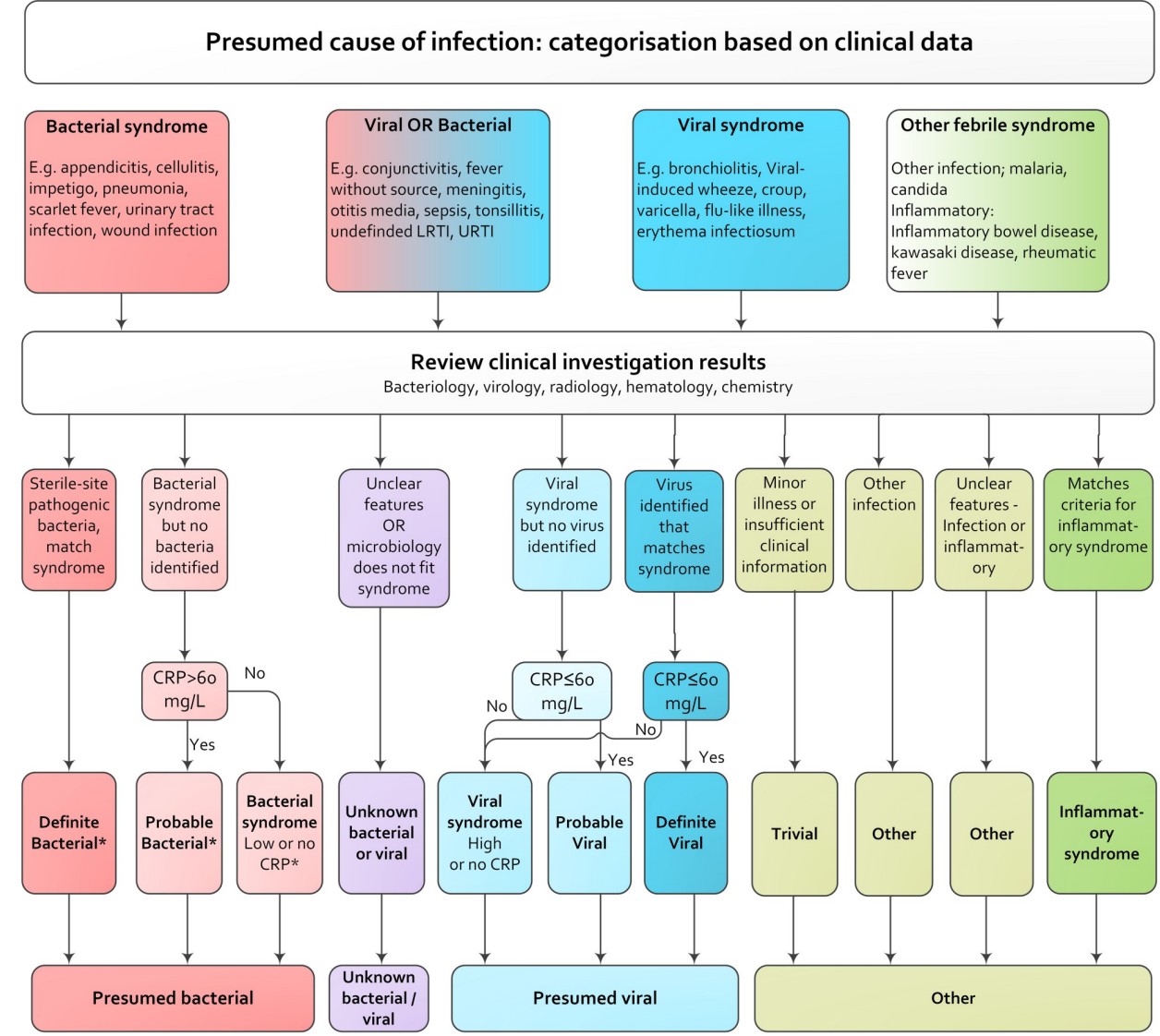

**Fig 1. Categorisation of presumed cause of infection.** CRP, C-reactive protein; LRTI, lower respiratory tract infection; URTI, upper respiratory tract infection.
*Patients could have identified viral co-infection.

We aimed to improve data quality and standardised data collection by using a training module for the local clinical and research teams to optimise clinical assessment and data collection for febrile children. This training module included clarification of the individual alarming signs and classification examples of common diagnoses. Furthermore, entry guidelines for the eCRF were available, monthly teleconferences and biannual meetings were organised, and quarterly reports of data quality for each ED were discussed. These consortium teleconferences also included discussion of difficult cases.

## Antibiotic classification

Antibiotics were categorised using the Anatomical Therapeutic Chemical classification including beta-lactamase sensitive penicillins (J01CE); beta-lactamase resistant penicillins (J01CF); penicillins with extended spectrum (J01CA); combinations of penicillins including beta-lactamase inhibitors (J01CR); macrolides (J01FA); first-generation, second-generation, and third-generation cephalosporins (J01DB, J01DC, J01DD); trimethoprim and sulphonamides (J01EA01, J01EE01); aminoglycosides (J01GB); quinolones (J01MA); glycopeptides (J01XA); and other antibiotics.

In addition, we compared the prescription of narrow-spectrum and broad-spectrum antibiotics. We explored the definitions reported in previous studies on antibiotic classification and used an expert opinion panel including paediatric infectious disease specialists and general paediatricians (PERFORM partners), to establish the final classification into broad-spectrum and narrow-spectrum for all systemic antibiotics [5,6,11,15,27,28]. Narrow-spectrum antibiotics comprised penicillins (e.g., amoxicillin) and first-generation cephalosporins. Broad-spectrum antibiotics included penicillins with beta-lactamase inhibitor combinations (e.g., amoxicillin/clavulanic acid), macrolides, aminoglycosides, glycopeptides, and second-generation and third-generation cephalosporins. Prescriptions of both broad-spectrum and narrow-spectrum antibiotics in the same patient were considered broad-spectrum. Topical antibiotics were not included. Details of this classification are presented in S5 Text.

## Outcomes

We assessed various aspects of antibiotic prescription: (1) antibiotic prescription rate; (2) the proportion of antibiotics that were broad-spectrum versus narrow-spectrum; (3) the proportion of antibiotics of 'likely appropriate' indication (presumed bacterial), 'likely inappropriate' indication (presumed viral), or 'inconclusive' indication (unknown bacterial/viral); (4) the proportion of oral antibiotics of inappropriate duration; and (5) the proportion of oral antibiotics that matched the antibiotic type in the local guideline ('guideline-concordant') in uncomplicated urinary and upper and lower RTIs. Antibiotic prescriptions were classified as likely appropriate in presumed bacterial infections (definite bacterial, probable bacterial, bacterial syndrome), likely inappropriate in presumed viral infections (definite viral, probable viral, viral syndrome), and inconclusive in unknown bacterial/viral infections or other infections. Inappropriate duration was defined as >10 days for treatment of tonsillitis with beta-lactamase sensitive penicillins (J01CE) and >7 days for all other prescriptions according to recommendations by international guidelines [29–31]. In addition, guideline-concordant prescription in patients with uncomplicated RTIs and uncomplicated urinary tract infections was defined according to the local guideline (S6 Text). Uncomplicated infections were defined as infections in previously healthy children who did not receive therapeutic antibiotic treatment before the ED visit.

## Data analysis

Missing values were assumed to be missing at random, and therefore we used multiple imputation by chained equations with the MICE package in R for the regression analysis. We

excluded patients with missing data on antibiotic prescription, presumed cause of infection, and focus of infection [32]. Only the first visit was included for patients who visited the ED again within 5 days.

First, we performed a descriptive analysis of the frequency of antibiotic prescription and broad-spectrum and narrow-spectrum prescription, including ranges across EDs. For all outcomes, we calculated the overall proportion and proportion per ED. Second, we used multilevel logistic regression with a random intercept for each ED to study variation between EDs in antibiotic prescription, broad-spectrum prescription versus narrow-spectrum prescription, and intravenous/intramuscular versus oral prescriptions [33]. In an adjusted model we corrected for patient-level factors and for hospital-level factors influencing antibiotic prescribing. Patient-level factors were selected a priori according to the literature [3,4,23,34,35] and included general characteristics (age, sex, season, comorbidity, referral [referred versus self-referred]), markers for disease severity such as triage urgency (high urgency [immediate, very urgent, urgent] versus low urgency [standard, non-urgent]), fever duration in days, fever measured at ED visit (≥38˚C), and presence of NICE guideline "red traffic light" alarming signs (0, 1, ≥2). We investigated diagnostics, including CRP (not performed or <20, 20–60, or >60 mg/l) [25,36], chest X-ray (not performed, normal, abnormal), and urinalysis (not performed, normal, abnormal [positive for leukocyte esterase and/or nitrite]). Furthermore, we included focus of infection (upper respiratory tract, lower respiratory tract, gastrointestinal, urinary tract, undifferentiated fever, skin/musculoskeletal, sepsis/central nervous system, flu-like illness/childhood exanthem, inflammatory/other) and diagnostic groups according to cause as classified by the flowchart in Fig 1: presumed bacterial (definite bacterial, probable bacterial, bacterial syndrome), unknown bacterial/viral, presumed viral (definite viral, probable viral, viral syndrome), and other.

For the hospital-level factors, we explored variables that varied between hospitals and were related to antibiotic prescribing [19,37–40]: total number of ED visits, supervision, availability of point-of-care tests (streptococcal antigen test and CRP), and primary care during out-of-office hours. We included hospital-level factors if they improved the model using univariate analysis. Linearity of continuous variables was tested using restricted cubic splines. Specifications of the adjusted model are presented in Table 1 and in S7 Text.

**Table 1. Variables in the adjusted model.**

| Category | Variables |
|---|---|
| **Patient-level factors** | |
| General characteristics | Age[*], sex, season, comorbidity, referral |
| Disease severity | Triage urgency, fever duration, fever measured at ED, presence of NICE alarming signs, previous antibiotic use[˟] |
| Diagnostics | C-reactive protein, chest X-ray, urinalysis |
| Infection | Focus of infection, cause of infection |
| **Hospital-level factors[±]** | Total number of ED visits, supervision, availability of point-of-care tests (streptococcal antigen test and C-reactive protein), primary care during out-of-office hours[±] |

[*]Age was modelled using restricted cubic splines (3 knots).

[˟]Previous antibiotic use was added in the models with outcome broad-spectrum versus narrow-spectrum prescription.

[±]None of the hospital-level factors were significant, and therefore they were not included in the final model.

ED, emergency department; NICE, National Institute for Health and Care Excellence.

Variation in antibiotic prescription rates between EDs was determined by 2 measures: standardised prescription rates and median odds ratios (MORs). We calculated standardised antibiotic prescription rates using indirect standardisation, where the expected number of antibiotic prescriptions was standardised to the average ED. Standardised antibiotic prescription ratios are the ratio between observed antibiotic prescriptions in an ED and the expected antibiotic prescriptions in an ED. The expected number of antibiotic prescriptions was estimated through the adjusted model, by summing the predicted probabilities from the adjusted model of antibiotic prescription for each of the patients. Standardised rates > 1 indicate higher prescription rates than expected, and standardized rates < 1 indicate lower prescription rates than expected. We visualised standardised rates in a heat map.

The MOR is a measure of variation between high- and low-prescribing clusters of EDs. The MOR reflects the difference in probability of receiving antibiotics comparing similar patients attending an ED with high antibiotic prescribing and an ED with low antibiotic prescribing. If the MOR is equal to 1.00, there is no variation between clusters, and if the MOR is high, this indicates important between-cluster variation [41,42].

Stratified analyses were performed in patients with and without comorbidities. Also, since antimicrobial resistance patterns vary greatly between European countries, standardised rates of broad-spectrum versus narrow-spectrum antibiotic prescription were compared with antimicrobial resistance data of invasive isolates on a national level and at the hospital level [11,43] (S8 Text). Correlations were calculated using the 2-tailed Spearman's rank coefficient ($\rho$). A $p$-value below 0.05 was considered significant.

### Appropriateness of antibiotic prescriptions

We calculated standardised rates for antibiotic prescription and broad-spectrum prescription in groups of presumed viral infections, presumed bacterial infections, and unknown bacterial/viral infections. Next, we assessed the proportion of all antibiotic prescriptions that were likely appropriate, likely inappropriate, and inconclusive. For all oral prescriptions, we calculated the proportion of prescriptions that were both inappropriate in indication (likely inappropriate) and of inappropriate duration, and the proportion of prescriptions that were either inappropriate in indication or inappropriate in duration. In uncomplicated RTIs and urinary tract infections, we calculated the proportion of all oral prescriptions that were inappropriate for all the 3 measures (indication, duration, and guideline concordance), and the proportion of prescriptions that were inappropriate in any of the 3 measures. R version 3.4 was used for the analysis and visualisation of the data.

## Results

### Study population

Of the total population of 38,480 patients, we excluded 738 patients based on missing data of antibiotics or diagnosis, and the repeated visit of 2,092 patients to the same ED. Compared to patients with complete outcome data, patients with missing data were similar in age, sex, comorbidity, and admission rate (S9 Text). In addition, there were no differences in completeness of outcomes and diagnosis between discharged and admitted patients.

For the analysis, we included 35,650 febrile children (median age 2.8 years [IQR 1.3–5.6], 54.6% male). The different EDs varied substantially in patients who were referred (range: 4.9%–99.2%), were ill appearing (range: 0.8%–47.4%), or had any comorbidity (range: 5.1%–65.3%) (Table 2). The most common infections were upper respiratory tract ($n$ = 18,783, 52.7%), lower respiratory tract ($n$ = 5,167, 14.5%), gastrointestinal tract ($n$ = 3,694, 10.4%), and undifferentiated fever ($n$ = 2,784, 7.8%). The incidence of sepsis and central nervous system

Table 2. Patient characteristics of the study population (n = 35,650).

| Characteristic | n (%) or median (IQR) | Range across EDs (%) | Missing, n (%) |
|---|---|---|---|
| **Age in years** | 2.77 (1.32–5.59) | | |
| **Male** | 19,476 (54.6) | 51.5–59.1 | 1 (0.0) |
| **Comorbidity** | 5,889 (16.5) | 5.1–65.3 | 326 (0.9) |
| **Season** | | | 1,111 (3.1) |
| Winter | 12,665 (35.5) | 26.8–53.2 | |
| Spring | 9,054 (25.4) | 18.2–31.2 | |
| Summer | 5,767 (16.2) | 9.5–23.5 | |
| Autumn | 8,164 (22.9) | 6.9–31.4 | |
| **Triage urgency** | | | 1,059 (2.9) |
| High: immediate, very urgent, urgent | 12,251 (34.4) | 8.3–88.5 | |
| Low: standard, non-urgent | 22,340 (62.7) | 10.1–91.6 | |
| **Referred** | 15,104 (42.4) | 4.9–99.2 | 1,110 (3.1) |
| **Fever duration in days** | 1.5 (0–3) | | 2,449 (6.9) |
| **NICE "red traffic light" alarming signs** | | | |
| Ill appearance | 5,567 (15.6) | 0.8–47.4 | 1,525 (4.3) |
| Work of breathing | 2,987 (8.4) | 3.2–25.7 | 4,482 (12.6) |
| Dehydration | 1,763 (4.9) | 0.4–15.2 | 6,323 (17.7) |
| Rash: petechiae/non-blanching | 1,039 (2.9) | 1.4–5.8 | 3,963 (11.1) |
| Decreased consciousness | 188 (0.5) | 0.1–3.8 | 334 (0.9) |
| Meningeal signs | 132 (0.4) | 0.1–1.7 | 1,807 (5.1) |
| Focal neurology | 121 (0.3) | 0.0–2.6 | 2,224 (6.2) |
| Status epilepticus | 60 (0.2) | 0.0–1.9 | 1,099 (3.1) |
| **C-reactive protein (CRP)** | | | |
| No CRP performed | 19,578 (54.9) | 7.9–93.2 | |
| <20 mg/l | 8,729 (24.5) | 3.2–58.4 | |
| 20–60 mg/l | 4,191 (11.8) | 1.9–24.9 | |
| >60 mg/l | 3,152 (8.8) | 1.6–30.2 | |
| **Chest X-ray** | | | |
| No | 30,662 (86.0) | 78.6–93.8 | |
| Normal | 1,931 (5.4) | 0.9–10.0 | |
| Abnormal | 3,057 (8.6) | 2.9–12.8 | |
| **Urinalysis** | | | |
| No | 26,691 (74.9) | 60.8–91.4 | |
| Normal | 7,210 (20.2) | 7.1–29.8 | |
| Abnormal | 1,749 (4.9) | 1.5–9.5 | |

ED, emergency department; IQR, interquartile range; NICE, National Institute for Health and Care Excellence.

infections was low (n = 270, 0.8%). The majority of the children had a presumed viral infection (n = 20,383, 57.2%); presumed bacterial infections occurred in 22.1% of the patients (definite bacterial, 4.1%; probable bacterial/bacterial syndrome, 18.1%), and unknown bacterial/viral infections in 14.6% (n = 5,200) (Table 3).

## Overall antibiotic prescriptions

The overall antibiotic prescription rate was 31.9% (n = 11,371), of which 67.2% (7,636/11,731) were oral administrations and 31.3% (3,564/11,371) were administered intravenously or intramuscularly (153 children received a single dose at the ED). One-third of patients were treated

Table 3. Patient characteristics of the study population: Outcomes, *n* = 35,650.

| Outcome | *n* (%) or median (IQR) | Range across EDs (%) |
|---|---|---|
| **Therapeutic antibiotics use in last 7 days**[*] | 3,592 (10.1) | 6.6–15.6 |
| **Antibiotic treatment duration, days** | 7 (5–10) | |
| **Antibiotics prescribed at ED visit or first day of hospital admission**[*] | 11,371 (31.9) | 22.4–41.6 |
| Narrow-spectrum | 5,401 (15.2) | 3.1–23.2 |
| Broad-spectrum | 5,887 (16.5) | 9.5–34.7 |
| **Antibiotic administration**[*] | | |
| Oral | 7,636 (21.4) | 10.4–34.2 |
| Intravenous/intramuscular | 3,564 (9.9) | 1.7–21.3 |
| **Admission**[*] | 9,000 (25.2) | 4.5–54.2 |
| **ICU admission**[*] | 147 (0.4) | 0.1–4.3 |
| **Focus of infection** | | |
| Upper respiratory tract | 18,783 (52.7) | 25.7–70.0 |
| Lower respiratory tract | 5,167 (14.5) | 8.5–26.4 |
| Gastrointestinal/surgical abdomen | 3,694 (10.4) | 6.0–19.2 |
| Undifferentiated fever | 2,784 (7.8) | 1.8–18.8 |
| Flu-like illness/exanthem | 1,753 (4.9) | 2.0–11.9 |
| Urinary tract | 1,231 (3.5) | 1.2–5.8 |
| Soft tissue/musculoskeletal | 876 (2.5) | 0.5–6.8 |
| Sepsis/central nervous system | 270 (0.8) | 0.0–3.9 |
| Inflammatory | 136 (0.4) | 0.0–1.3 |
| Other | 957 (2.7) | 1.2–8.4 |
| **Cause of infection** | | |
| Presumed viral | 20,383 (57.2) | 37.3–71.4 |
| Definite bacterial | 1,451 (4.1) | 1.6–10.9 |
| Probable bacterial/bacterial syndrome | 6,438 (18.1) | 4.7–31.8 |
| Unknown bacterial/viral | 5,200 (14.6) | 1.6–37.9 |
| Other | 2,178 (6.1) | 1.1–30.9 |

[*]Missing: therapeutic antibiotic use in last 7 days, 681/35,650 (1.9%); antibiotic duration, 1,980/11,371 (17.4%); broad-spectrum versus narrow-spectrum antibiotics, 83/11,371 (0.7%); antibiotic administration, 171/11,371 (1.5%); admission and ICU admission, 25/35,650 (0.1%).

ED, emergency department; ICU, intensive care unit; IQR, interquartile range.

with antibiotics for over 7 days (3,534/9,391, 37.6%) (S1 Fig). The types of antibiotics most often prescribed were penicillins with extended spectrum (3,220/11,371, 28.3%), combinations of penicillins with beta-lactamase inhibitors (2,309/11,371, 20.3%), and beta-lactamase sensitive penicillins (2,001/11,371, 17.6%). Half of the prescribed antibiotics were broad-spectrum agents (5,887/11,371, 51.7%). The most prescribed broad-spectrum antibiotics were combinations of penicillins with beta-lactamase inhibitors (2,309/11,371, 20.3%), second-generation cephalosporins (1,154/11,371, 10.1%), and third-generation cephalosporins (1,097/11,371, 9.6%) (Table 4; Fig 2).

## Variation of overall antibiotic prescription and broad-spectrum prescription

The proportion of febrile children receiving an antibiotic prescription ranged from 22.4% to 41.6% across EDs, and the proportion of those prescriptions that were for broad-spectrum

**Table 4. Frequencies of antibiotic classes and ranges across EDs (*n* = 11,371).**

| Antibiotic class | *n* (%) | Range across EDs (%) |
|---|---|---|
| Beta-lactamase sensitive penicillins (e.g., benzylpenicillin) | 2,001 (17.6) | 0.1–32.5 |
| Beta-lactamase resistant penicillins (e.g., flucloxacillin) | 167 (1.5) | 0.0–8.1 |
| Penicillins with extended spectrum (e.g., amoxicillin) | 3,220 (28.3) | 2.6–61.6 |
| Combinations of penicillins with beta-lactamase inhibitors (e.g., amoxicillin with clavulanate) | 2,309 (20.3) | 1.4–59.0 |
| Macrolides (e.g., azithromycin) | 638 (5.6) | 2.9–11.0 |
| First-generation cephalosporins | 167 (1.4) | 0.0–9.8 |
| Second-generation cephalosporins | 1,154 (10.1) | 0.0–25.6 |
| Third-generation cephalosporins | 1,097 (9.6) | 1.1–25.1 |
| Trimethoprim and sulphonamides | 128 (1.1) | 0.0–5.1 |
| Aminoglycosides | 205 (1.8) | 0.0–15.6 |
| Quinolones | 51 (0.4) | 0.0–2.8 |
| Glycopeptides | 31 (0.3) | 0.0–2.7 |
| Other | 120 (1.1) | 0.0–4.6 |
| *Missing* | 83 (0.7) | 0.0–3.6 |

ED, emergency department.

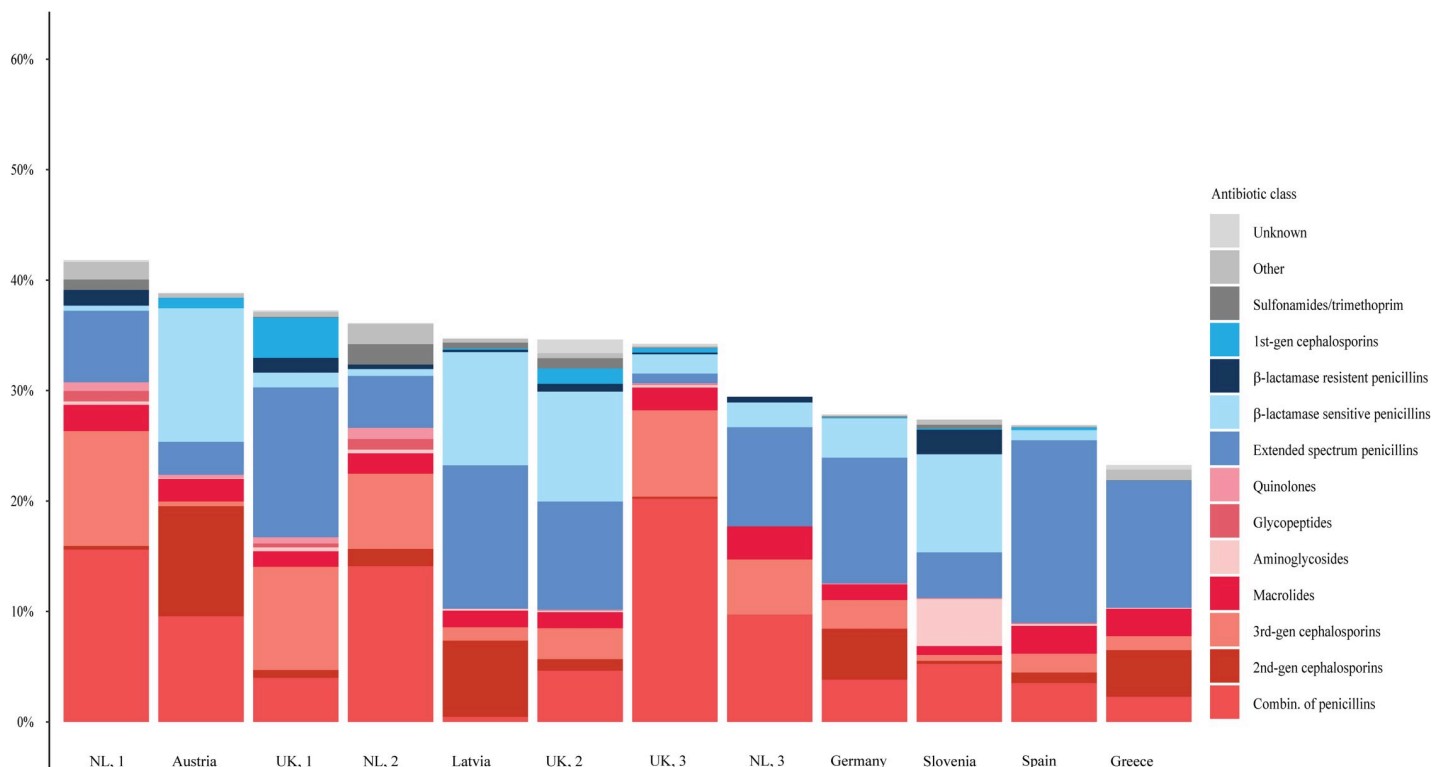

**Fig 2. Antibiotic classes of prescribed antibiotics across EDs, *n* = 35,650.** Red shades indicate broad-spectrum classes, blue shades indicate narrow-spectrum classes, and grey shades indicate unclassified classes and prescriptions of unknown class. EDs are sorted by antibiotic prescription rate. ED, emergency department; NL, the Netherlands; UK, United Kingdom.

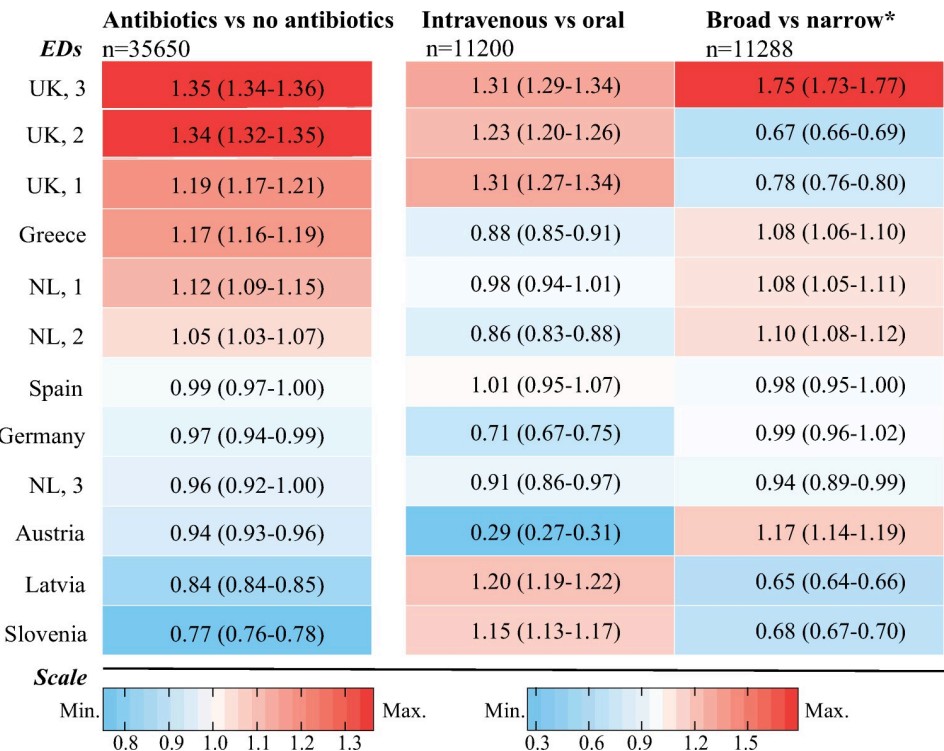

**Fig 3. Heat map of standardised prescription rates by ED (95% CI).** All adjusted for age, sex, season, comorbidity, referral, triage urgency, fever measured at ED, fever duration, alarming signs, CRP, chest X-ray, urinalysis, focus of infection, and cause of infection. EDs are ordered according to standardised antibiotic prescribing rate, from low to high on the left vertical axis. The coloured boxes represent rank of standardised rate for each ED: Red indicates rates > 1, blue indicates rates < 1, and rates equal to 1 are white. *Also adjusted for previous antibiotic use. ED, emergency department; NL, the Netherlands; UK, United Kingdom.

agents ranged from 33.0% to 90.3%. Of the broad-spectrum agents, penicillins with beta-lactamase inhibitors had the largest variation (range 1.4%–59.0%), but other broad-spectrum agents varied as well (range 17.3%–37.0%). Fig 3 presents the standardised prescription rates from the adjusted model. None of the hospital-level factors was related with antibiotic prescription (*p*-value range: 0.14–0.77). After correction for general patient characteristics (age, sex, season, comorbidity, referral), disease severity (triage urgency, fever duration, fever measured at ED, alarming signs), diagnostics, focus of infection, and cause of infection, variability of antibiotic prescriptions remained between EDs in the adjusted model (range of standardised prescription rates: 0.77–1.35; MOR 2.41). Variation was also observed for intravenous versus oral administration (range of standardised rates: 0.29–1.31; MOR 2.60) and prescription of broad-spectrum antibiotics versus narrow-spectrum antibiotics (range of standardised rates: 0.65–1.75; MOR 3.20). Stratified for comorbidity, standardised antibiotic prescription rates and broad-spectrum rates were comparable in children with and without comorbidity. Higher standardised rates for broad-spectrum antibiotics were not related to higher antimicrobial resistance percentages on a national level or on a hospital level (S8 Text). Results of variation of antibiotic and broad-spectrum prescriptions for RTIs are provided in S10 Text.

## Variation of antibiotic and broad-spectrum prescriptions in viral infections, bacterial infections, and unknown bacterial/viral infections

The antibiotic prescription rate was 6.9% (1,418/20,383) for presumed viral infections (range across EDs: 0.4%–18.9%), 88.8% (1,289/1,451) for definite bacterial infections (range across

EDs: 83.5%–96.2%), 94.7% (6,097/6,438) for probable bacterial/bacterial syndrome infections (range across EDs: 81.2%–99.3%), and 45.2% (2,348/5,200) for unknown bacterial/viral infections (range across EDs: 1.7%–79.3%) (S2 Fig).

Adjusted for general characteristics, disease severity, diagnostics, and focus of infection, we observed variation for antibiotic prescriptions in presumed viral infections (range of standardised rates: 0.05–3.29; MOR 4.91) and unknown bacterial/viral infections (range of standardised rates: 0.05–1.78; MOR 4.78) (Fig 4). Antibiotic prescriptions varied less for patients with presumed bacterial infections (range of standardised rates: 0.91–1.06; MOR 2.32). The proportion of broad-spectrum prescriptions was 74.1% (1,037/1,399) for presumed viral infections (range across EDs: 38.9%–91.4%), 68.5% (880/1,284) for definite bacterial infections (range across EDs: 39.2%–96.0%), 43.2% for probable bacterial/bacterial syndrome infections (2,628/6,081, range across EDs: 28.5%–86.3%), and 51.6% (1,191/2,306) for unknown bacterial/viral infections (range across EDs: 20.0%–95.7%) (S2 Fig). After adjustment, differences for broad-spectrum versus narrow-spectrum antibiotics remained for presumed viral infections (range of standardised rates: 0.57–1.54; MOR 2.59), presumed bacterial infections (range of standardised rates: 0.66–1.86; MOR 3.09), and unknown bacterial/viral infections (range of standardised rates: 0.44–1.64; MOR 3.70) (S3 Fig).

## Variation in prescriptions of appropriate indication and appropriate duration

Of all antibiotic prescriptions, 65.0% (7,386/11,371) were determined to be likely appropriate (range across EDs: 23.7%–98.9%), 12.5% (1,418/11,371) were likely inappropriate (range

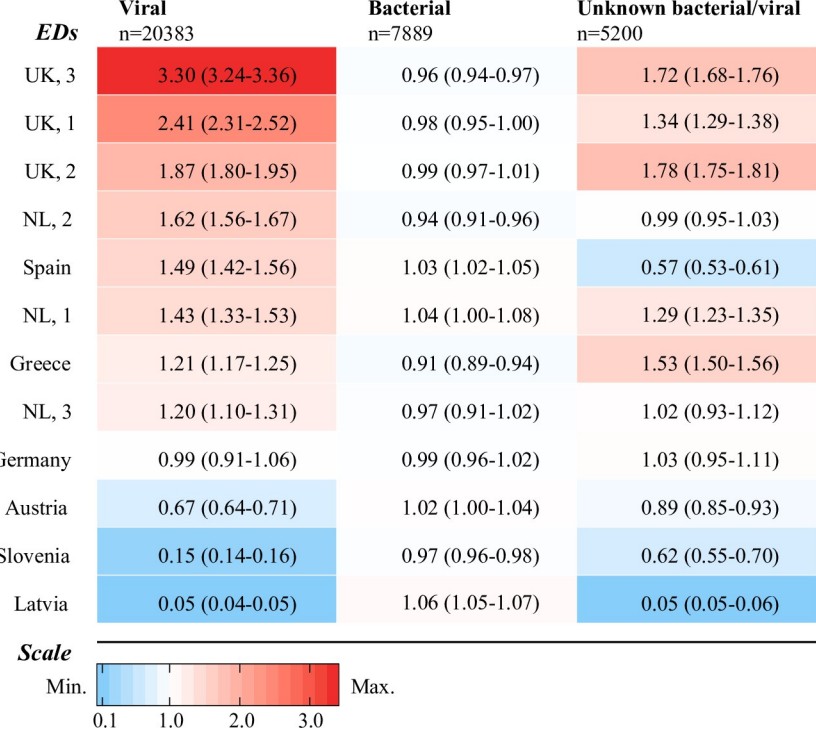

| EDs | Viral n=20383 | Bacterial n=7889 | Unknown bacterial/viral n=5200 |
|---|---|---|---|
| UK, 3 | 3.30 (3.24-3.36) | 0.96 (0.94-0.97) | 1.72 (1.68-1.76) |
| UK, 1 | 2.41 (2.31-2.52) | 0.98 (0.95-1.00) | 1.34 (1.29-1.38) |
| UK, 2 | 1.87 (1.80-1.95) | 0.99 (0.97-1.01) | 1.78 (1.75-1.81) |
| NL, 2 | 1.62 (1.56-1.67) | 0.94 (0.91-0.96) | 0.99 (0.95-1.03) |
| Spain | 1.49 (1.42-1.56) | 1.03 (1.02-1.05) | 0.57 (0.53-0.61) |
| NL, 1 | 1.43 (1.33-1.53) | 1.04 (1.00-1.08) | 1.29 (1.23-1.35) |
| Greece | 1.21 (1.17-1.25) | 0.91 (0.89-0.94) | 1.53 (1.50-1.56) |
| NL, 3 | 1.20 (1.10-1.31) | 0.97 (0.91-1.02) | 1.02 (0.93-1.12) |
| Germany | 0.99 (0.91-1.06) | 0.99 (0.96-1.02) | 1.03 (0.95-1.11) |
| Austria | 0.67 (0.64-0.71) | 1.02 (1.00-1.04) | 0.89 (0.85-0.93) |
| Slovenia | 0.15 (0.14-0.16) | 0.97 (0.96-0.98) | 0.62 (0.55-0.70) |
| Latvia | 0.05 (0.04-0.05) | 1.06 (1.05-1.07) | 0.05 (0.05-0.06) |

Scale

Min. ▭ Max.

0.1   1.0   2.0   3.0

**Fig 4. Heat map of standardised antibiotic prescription rates by ED for presumed viral, presumed bacterial, and unknown bacterial/viral infections (95% CI).** All adjusted for age, sex, season, comorbidity, referral, triage urgency, fever duration, alarming signs, CRP, chest X-ray, urinalysis, and focus of infection. EDs are ordered according to standardised antibiotic prescribing rate, from low to high on the left vertical axis. The coloured boxes represent rank of standardised rate for each ED: Red indicates rates > 1, blue indicates rates < 1, and rates equal to 1 are white. ED, emergency department; NL, the Netherlands; UK, United Kingdom.

across EDs: 0.6%–29.3%), and 22.6% (2,567/11,371) were inconclusive (range across EDs: 0.5%–61.7%).

Oral antibiotic prescriptions with inappropriate duration were found in 20.0% (1,525/7,636) of prescriptions, and this ranged from 4.4% to 59.0% across EDs (Fig 5). Of all oral antibiotic prescriptions, 2.1% (134/7,636) were of both inappropriate indication and inappropriate duration (range across EDs: 0.0%–8.4%), whereas 30.0% (2,294/7,636) were either of inappropriate indication or of inappropriate duration (range across EDs: 11.3%–69.9%).

## Variation of appropriate prescriptions in uncomplicated RTIs and urinary tract infections

In uncomplicated RTIs, oral prescriptions were not guideline-concordant in 22.3% (973/4,373) of prescriptions (range across EDs: 11.8%–47.3%) (Fig 5). In this group, the proportion

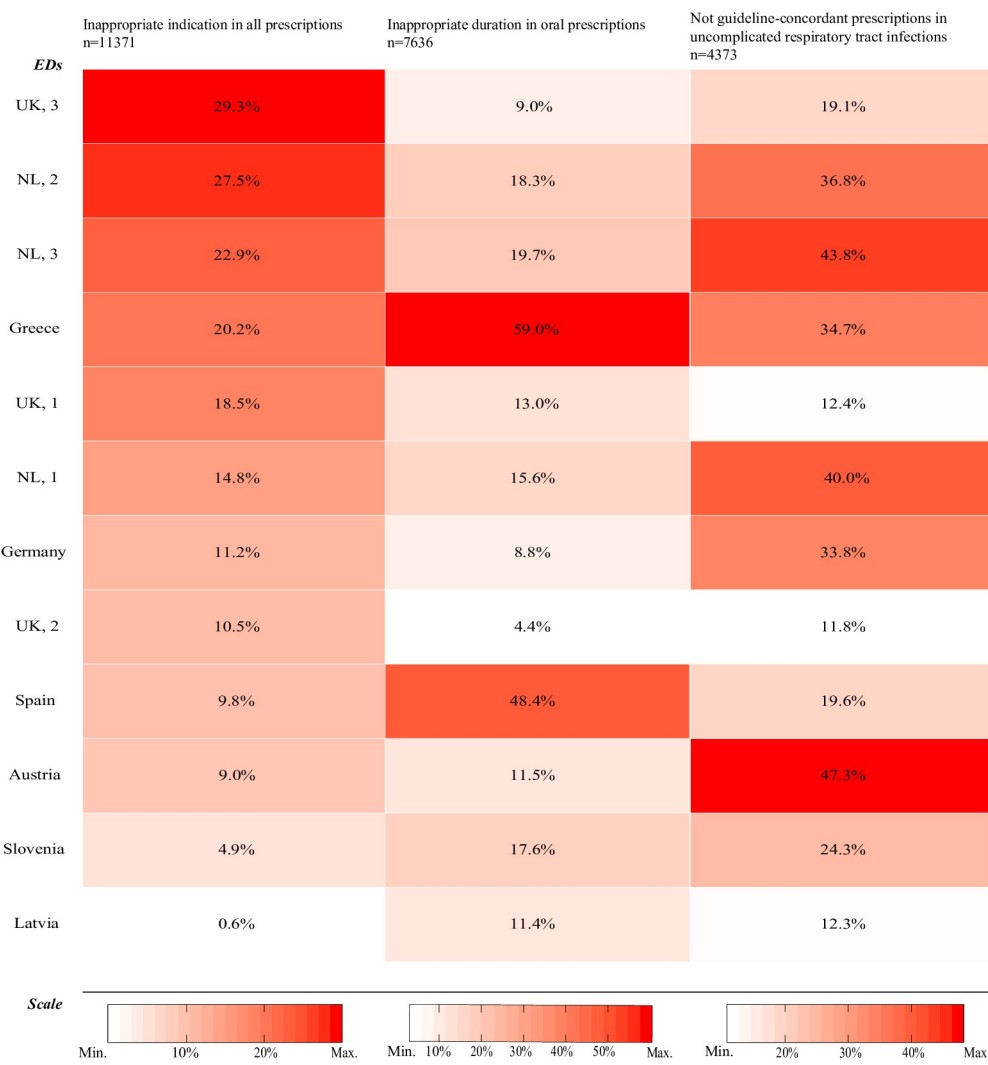

**Fig 5. Heat map of inappropriateness of antibiotic prescriptions across EDs.** EDs are ordered according to proportion of inappropriately indicated prescriptions, from low to high on the left vertical axis. The coloured boxes represent rank of proportion for each ED: Red indicates the highest proportion, and white indicates the lowest proportion. ED, emergency department; NL, the Netherlands; UK, United Kingdom.

of prescriptions that were inappropriate in all 3 measures (indication, duration, and guideline concordance) was 0.7% (31/4,373), whilst 42.3% (1,850/4,373) were inappropriate in any of the 3 measures (range across EDs: 15.7%–80.9%). In uncomplicated urinary tract infections, oral prescriptions were not concordant with the local guideline in 45.1% of prescriptions (152/337) (range across EDs: 11.1%–100%), and 65.9% (222/337) were inappropriate in any of the 3 measures (range across EDs: 11.1%–100%).

## Discussion

In this large prospective multicentre study, we found diversity in antibiotic prescriptions, and in particular broad-spectrum antibiotic prescriptions, for febrile children attending different EDs in Europe. After adjustment for general characteristics, disease severity, diagnostics, and focus of infection, we observed minor variation in antibiotic prescriptions for bacterial infections, and larger variability in antibiotic prescriptions for viral infections and unknown bacterial/viral infections. Moreover, one-third of all antibiotic prescriptions were of inappropriate or inconclusive indication, and 20% of oral prescriptions were of inappropriate duration, with large variation across EDs. Between EDs, the proportion of oral prescriptions that were not concordant with the local guideline varied from 12% to 47% in RTIs and from 11% to 100% in urinary tract infections.

Our study supports previous studies that reported variable antibiotic prescribing for all febrile children, but found less variation than a previous study in children with RTIs across 28 European EDs (range of standardised rates: 0.5–2.0) [5]. In contrast to this study, our study corrected for aetiology of infection—bacterial, viral, or unknown—based on a standardised flowchart. Studies in the US on diversity in outpatient antibiotic prescribing found regional differences in both antibiotic and broad-spectrum prescribing [6,10,17]. These studies, however, were not focused on ED visits alone since all ambulatory visits were included.

The Access, Watch, and Reserve (AWaRE) classification has recently been used to classify global antibiotic prescriptions in 2 studies assessing oral formulations and use of inpatient antibiotics in children [14,44,45]. These studies confirmed variable patterns of antibiotic prescribing between countries, but did not adjust for differences in population and did not report data of emergency care visits. Further, the AWaRE classification led to a substantial proportion of unclassified antibiotics in our study population (12.2%; range across EDs: 1.9%–26.9%) and absence of the reserve category.

Previous studies in the US have evaluated appropriateness of antibiotic prescribing in children defined by ICD codes. Poole et al. [46] found that prescriptions were in general not indicated in 32% of emergency care visits in children. Additionally, overall prescription of first-line antibiotics (amoxicillin, amoxicillin-clavulanate) ranged from 50% to 78% for RTIs in children [46–48]. We found a similar rate of guideline-concordant prescriptions in RTIs (78%), whilst guideline concordance was defined differently for most EDs: amoxicillin and narrow-spectrum penicillins according to the local guideline. One ED (UK, 3) used amoxicillin-clavulanate as first-line for RTIs. Our study is the first to our knowledge to evaluate appropriateness of antibiotic prescribing in febrile children visiting different EDs in Europe, using a structured flowchart categorising viral, bacterial, and unclassified infections, and taking local guidelines into account.

Strengths of this European multicentre study include the large sample size, detailed patient information, recruitment in a diverse range of ED settings in 8 EU countries, and recruitment over a full year to reflect seasonal variation. Furthermore, a rigorous, standardised structured assessment of all cases was carried out to establish the presumed cause of infection, using a consensus-based flowchart taking into account clinical syndrome, CRP, and culture results.

Previous studies have addressed appropriate prescribing for diagnoses based on ICD codes. This classification, however, may not accurately take into account bacterial or viral aetiology [6,10,17,46]. Our large sample size enabled adjustment for hospital- and patient-level factors influencing antibiotic use in the EDs [19].

This study has some limitations. First, the included EDs are not representative of all febrile children attending the ED in that country. The EDs participating in this study are university hospitals or large teaching centres with intensive care unit facilities involved in paediatric infectious disease research collaborations. Fever and sepsis guidelines were available in all EDs [19]. Therefore, these EDs represent a high standard of care, and generalisation of our findings to smaller hospitals or to a regional or national level should be undertaken with caution. However, we corrected for the most important confounders including comorbidities, multiple markers of disease severity, and focus and presumed cause of infection. Second, although the experience of the physician (resident or consultant) and clinician specialty are related with antibiotic prescription [49,50], we could not adjust for physician background at the patient level. However, we evaluated the contribution of supervision to antibiotic prescription at the hospital level. In our study, supervision was not related to antibiotic prescription. Our efforts to improve data quality by training clinical and research staff might have influenced common clinical practice. Since this training focused on awareness of alarming signs in the clinical assessment of the febrile child, it is unlikely that it influenced antibiotic prescription. Furthermore, we only included the first visit of patients who repeatedly visited the ED, since data collection did not include secondary visits in all EDs.

Differences in antibiotic prescribing could be influenced by differences in immunisation coverage. In our study, countries with lower coverage for pneumococcal vaccinations (<90%) (Germany, Slovenia) did not have higher antibiotic prescriptions at the ED [19,51].

We found large variation in broad-spectrum prescriptions across the different EDs. Increased antimicrobial resistance rates could possibly explain higher broad-spectrum prescribing. We compared broad-spectrum rates with national data for antimicrobial resistance and hospital methicillin resistance rates. Interestingly, EDs based in countries with higher antimicrobial resistance on a population level (e.g., Greece, Spain) prescribed less broad-spectrum agents than expected in the ED. These hospitals with higher burden of national antimicrobial resistance may perceive more problems with antimicrobial resistance and might feel a greater pressure to reduce antibiotic prescriptions in the ED. It should be noted that antibiotic prescribing in the ED will not be representative of antibiotic prescription patterns of primary care in the community.

The diversity in antibiotic prescribing across different EDs appears not to be associated with antimicrobial resistance or immunisation coverage. Although the ideal antibiotic prescription rate is unknown, the diversity in antibiotic prescribing suggests overprescribing. Prescription rates were above the average incidence of serious bacterial infections. We found variation in antibiotic prescription rates, even when adjusting for general characteristics, disease severity, diagnostics, and focus and cause of infection.

This suggests room for improvement in reduction of antibiotic prescriptions and especially broad-spectrum prescriptions at the ED. EDs with higher antibiotic prescription rates did not necessarily prescribe more broad-spectrum antibiotics. The ED with the highest standardised broad-spectrum rate (UK, 3) did not have a high proportion of inappropriate prescriptions for RTIs. In only this ED, amoxicillin-clavulanate (broad-spectrum) was the first-choice agent for uncomplicated RTIs, which could explain the higher broad-spectrum rate in this ED. Studies demonstrated that use of narrow-spectrum antibiotics compared to broad-spectrum antibiotics leads to similar clinical outcomes and to fewer adverse events [28,52]. Unnecessary use of broad-spectrum antibiotics potentially increases resistance rates even further.

In addition, diversity of antibiotic prescription increased with diagnostic uncertainty. After adjustment for general characteristics, disease severity, diagnostics, and focus of infection, we observed minor variation in antibiotic prescriptions for bacterial infections, and larger variability in antibiotic prescriptions for viral infections and unknown bacterial/viral infections. In general, EDs with higher antibiotic prescription rates in viral infections also had higher antibiotic prescription rates in unknown bacterial/viral infections. This indicates that overprescribing in viral infections is linked to higher prescriptions in unknown bacterial/viral infections. Diagnostic uncertainty in patients with an unclear cause of infection could be reduced by improved targeted antibiotic prescription from new diagnostic signatures of bacterial and viral infection.

We evaluated appropriateness in indication, duration, and guideline concordance. Ideally, EDs should target 100% appropriateness in these 3 aspects of antibiotic prescribing. In our study, we did not observe a clear association between inappropriately indicated prescriptions and prescriptions of inappropriate duration. This indicates that guideline implementations should focus on these different aspects of appropriate antibiotic prescribing to ensure prescriptions of appropriate indication, duration, and antibiotic selection. Furthermore, quality improvement initiatives should be emphasised in EDs with higher proportions of inappropriate prescriptions. In addition, future antimicrobial stewardship interventions across Europe should focus on reducing broad-spectrum treatment and antibiotic use in viral infections.

To conclude, we found substantial variation in antibiotic prescriptions and especially broad-spectrum antibiotic prescriptions in European EDs after adjustment for patient characteristics, disease severity, diagnostics, and focus and cause of infection. The proportion of antibiotic prescriptions in bacterial infections was comparable between EDs, but diversity was especially large in antibiotic prescriptions for viral infections and unknown viral/bacterial infections. This variation indicates overprescription of antibiotics in these groups of patients. Furthermore, indications of prescriptions were inappropriate or inconclusive in one-third of prescriptions, and this proportion varied between EDs. In respiratory and urinary infections, guideline concordance of prescriptions varied widely across EDs. Until better diagnostics are available to accurately differentiate between bacterial and viral aetiologies, we strongly urge the implementation of antimicrobial stewardship guidelines to reduce antibiotic prescription in febrile children across Europe.

## Supporting information

**S1 Fig. Duration of prescribed antibiotics.**
(PDF)

**S2 Fig. Range of antibiotic prescriptions and broad-spectrum prescriptions by emergency department (ED) for viral, bacterial, and unknown bacterial/viral infections.** (A) antibiotic prescriptions; (B) broad-spectrum prescriptions.
(PDF)

**S3 Fig. Heat map of standardised broad-spectrum versus narrow-spectrum rates for viral, bacterial, or unknown bacterial/viral infections.**
(PDF)

**S1 Text. STROBE checklist.**
(PDF)

**S2 Text. Statistical analysis plan.**
(PDF)

**S3 Text. Ethics committees of participating hospitals.**
(PDF)

**S4 Text. Hospital characteristics.**
(PDF)

**S5 Text. Broad-spectrum and narrow-spectrum antibiotic definitions.**
(PDF)

**S6 Text. Local guidelines of antibiotic treatment.**
(PDF)

**S7 Text. Details of the adjusted model.**
(PDF)

**S8 Text. National and hospital antimicrobial resistance data.**
(PDF)

**S9 Text. Descriptive characteristics of cases with complete outcomes and cases with missing outcomes.**
(PDF)

**S10 Text. Variation of antibiotic and broad-spectrum prescription in lower RTIs and otitis media, tonsillitis/pharyngitis, and other upper RTIs.**
(PDF)

**S11 Text. PERFORM consortium.**
(PDF)

## Acknowledgments

We acknowledge all research nurses for their help in collecting data, and Anda Nagle (Riga) and the Institute of Microbiology at University Medical Centre Ljubljana for their help in collecting data on antimicrobial resistance. Members of the PERFORM consortium are listed in S11 Text.

## Author Contributions

**Conceptualization:** Nienke N. Hagedoorn, Dorine M. Borensztajn, Ruud Nijman, Anda Balode, Ulrich von Both, Enitan D. Carrol, Michiel van der Flier, Ronald de Groot, Benno Kohlmaier, Emma Lim, Ian Maconochie, Federico Martinon-Torres, Marko Pokorn, Franc Strle, Maria Tsolia, Shunmay Yeung, Dace Zavadska, Werner Zenz, Clementien Vermont, Michael Levin, Henriëtte A. Moll.

**Data curation:** Nienke N. Hagedoorn, Dorine M. Borensztajn, Ruud Nijman, Anda Balode, Ulrich von Both, Enitan D. Carrol, Irini Eleftheriou, Marieke Emonts, Michiel van der Flier, Ronald de Groot, Jethro Herberg, Benno Kohlmaier, Emma Lim, Ian Maconochie, Federico Martinon-Torres, Marko Pokorn, Franc Strle, Maria Tsolia, Dace Zavadska, Werner Zenz, Clementien Vermont, Michael Levin, Henriëtte A. Moll.

**Formal analysis:** Nienke N. Hagedoorn, Daan Nieboer, Henriëtte A. Moll.

**Methodology:** Nienke N. Hagedoorn, Dorine M. Borensztajn, Ruud Nijman, Ulrich von Both, Enitan D. Carrol, Marieke Emonts, Michiel van der Flier, Ronald de Groot, Jethro Herberg, Benno Kohlmaier, Emma Lim, Federico Martinon-Torres, Daan Nieboer, Marko Pokorn,

Maria Tsolia, Shunmay Yeung, Dace Zavadska, Clementien Vermont, Michael Levin, Henriëtte A. Moll.

**Supervision:** Clementien Vermont, Henriëtte A. Moll.

**Visualization:** Nienke N. Hagedoorn, Daan Nieboer.

**Writing – original draft:** Nienke N. Hagedoorn.

**Writing – review & editing:** Dorine M. Borensztajn, Ruud Nijman, Anda Balode, Ulrich von Both, Enitan D. Carrol, Irini Eleftheriou, Marieke Emonts, Michiel van der Flier, Ronald de Groot, Jethro Herberg, Benno Kohlmaier, Emma Lim, Ian Maconochie, Federico Martinon-Torres, Daan Nieboer, Marko Pokorn, Franc Strle, Maria Tsolia, Shunmay Yeung, Dace Zavadska, Werner Zenz, Clementien Vermont, Michael Levin, Henriëtte A. Moll.

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
