## [Decision Letter · Decision Letter 0]

13 Jan 2020

Dear Dr. Moll,

Thank you very much for submitting your manuscript "Variation in antibiotic prescription rates in febrile children presenting to Emergency Departments across Europe: PERFORM, an observational multicentre study" (PMEDICINE-D-19-04008) for consideration at PLOS Medicine. 

Your paper was evaluated by a senior editor, discussed with an academic editor with relevant expertise, and sent to independent reviewers, including a statistical reviewer. The reviews are appended at the bottom of this email and any accompanying reviewer attachments can be seen via the link below:

[LINK]

In light of these reviews, I am afraid that we will not be able to accept the manuscript for publication in the journal in its current form, but we would like to consider a revised version that addresses the reviewers' and editors' comments. Obviously we cannot make any decision about publication until we have seen the revised manuscript and your response, and we plan to seek re-review by one or more of the reviewers. 

We expect to receive your revised manuscript by Feb 03 2020 11:59PM. Please email us (plosmedicine@plos.org) if you have any questions or concerns.

We look forward to receiving your revised manuscript. 

Sincerely,

Louise Gaynor-Brook, MBBS PhD

Associate Editor 

PLOS Medicine

plosmedicine.org

General comments: Please replace ‘gender’ with ‘sex’ throughout the manuscript 

Please position reference brackets before any punctuation (comma or full stop), separated by a space.

Please revise your title according to PLOS Medicine's style. We suggest “Variation in antibiotic prescription rates in febrile children presenting to Emergency Departments across Europe (PERFORM): a multicentre observational study”

Data Availability: PLOS Medicine requires that the de-identified data underlying the specific results in a published article be made available, without restrictions on access, in a public repository or as Supporting Information at the time of article publication, provided it is legal and ethical to do so. Please see the policy at http://journals.plos.org/plosmedicine/s/data-availability and FAQs at

http://journals.plos.org/plosmedicine/s/data-availability#loc-faqs-for-data-policy

If the data are not freely available, please describe briefly the ethical, legal, or contractual restriction that prevents you from sharing it. Please also include an appropriate contact (web or email address) for inquiries (please note that this cannot be a study author).

Abstract Background: Provide expand upon the context of why the study is important. The final sentence should clearly state the study question. Please define ‘PERFORM’

Abstract Methods and Findings:

Please provide more detail on the setting (e.g. which European countries, types of hospital) and brief demographic details on the study population (e.g. age, sex)

In the last sentence of the Abstract Methods and Findings section, please describe the main limitation(s) of the study's methodology.

Please begin your Abstract Conclusions with "In this study, we observed ..." or similar.

Please interpret the study based on the results presented in the abstract, emphasizing what is new.

Please expand upon your Introduction to address past research and explain the need for and potential importance of your study. Indicate whether your study is novel and how you determined that. If there has been a systematic review of the evidence related to your study (or you have conducted one), please refer to and reference that review and indicate whether it supports the need for your study. 

Line 123 - please correct ‘ED’s’ to ‘EDs’

Methods 

Thank you providing a STROBE checklist. Please add the following statement, or similar, to the Methods: "This study is reported as per the Strengthening the Reporting of Observational Studies in Epidemiology (STROBE) guideline (S1 Checklist)." When completing the checklist, please use section and paragraph numbers, rather than page numbers.

Did your study have a prospective protocol or analysis plan? Please state this (either way) early in the Methods section. If a prospective analysis plan was used in designing the study, please include the relevant prospectively written document with your revised manuscript as a Supporting Information file to be published alongside your study, and cite it in the Methods section. A legend for this file should be included at the end of your manuscript. If no such document exists, please make sure that the Methods section transparently describes when analyses were planned, and when/why any data-driven changes to analyses took place. 

Results

Throughout your results, please indicate what the results in brackets represent (mean? and range)

Line 326-328: Please provide 95% CIs and p values

Lines 337 & 443 - Please indicate which factors are adjusted for 

Line 356 - please clarify what is meant by ‘prescription rates remained’

Line 359 - please correct to ‘upper respiratory tract infection’

At line 374, you describe the study as a "large prospective multicentre study". It may be that the study reports a retrospective analysis of prospectively collected data, and we ask you to adapt the language as appropriate. 

Discussion

Please remove subheadings within the Discussion section 

Please present and organize the Discussion as follows: a short, clear summary of the article's findings; what the study adds to existing research and where and why the results may differ from previous research; strengths and limitations of the study; implications and next steps for research, clinical practice, and/or public policy; one-paragraph conclusion.

References

Please ensure that journal titles are appropriately formatted and capitalised e.g. ref 7 BMJ; ref 22 JAMA; ref 43 formatting issues

Please add additional access details to reference 16.

Can the title of reference 43 be translated?

Supplementary Files

S2 Appendix - ‘investigation’ is misspelt in the the figure 

S3 Appendix - please correct ref 5

Comments from the reviewers:

Reviewer #1: This descriptive, cross-sectional study documents widespread variation in antibiotic prescribing among 12 European emergency departments to children who either present with fever or had a recent history of fever. The authors find widespread variation in prescribing particularly for viral conditions or conditions in which the etiology is either viral of bacterial. 

The authors have collected a large, rich, and impressive set of clinical data - by far the greatest strength of this paper. Moreover, there is no question that antibiotic stewardship in the ED for children is of utmost importance. The manuscript is generally easy to follow and the figures are helpful, particularly the Appendix figure with a flow chart with how the various categories like "presumed bacterial" were defined.

Despite these strengths, my major concern is that the research question the authors have asked - i.e., what is the degree of variation in antibiotic prescribing among these 12 European EDs - seems like the least interesting question they could have addressed using their impressive data. Based on the way it is framed, this study runs the risk of being seen as another paper documenting variation in care among some defined group of units (in this case, EDs). The weaknesses of such papers are 1) So many similar papers have been published that the only truly shocking finding would be the LACK of variation in care; and 2) It is difficult to know exactly to know what to do when variation is demonstrated, as the "appropriate" amount of variation is unknown even when you can adjust for the rich set of clinical variables that the authors have access to. This is the Achilles heel of variations-based analyses - at best, they can hint at potential overuse of care (in this case, antibiotics).

Some of the more important questions that could have been asked with these rich data include:

1) Why is there so much apparent over-testing? 45% of the children received a CRP - perhaps this is just a trans-Atlantic difference, but CRP is not routinely obtained for febrile children in the ED in the U.S., nor is it viewed as a particularly useful discriminator between viral and bacterial infections. Similarly, 25% of children received a urinalysis, but the focus of infection was the urinary tract for just 3.5% of children. These numbers boggle my mind. 

2) What is the estimated rate of antibiotic overuse? The authors could devise a system, for example, in which they consider antibiotics to be likely inappropriate if the patient had a confirmed or probable viral infection; likely appropriate if there was a confirmed or probable bacterial infection; and possibly appropriate for everything else. They could use their rich data on lab results, CXR results, and diagnoses to further their classification scheme. They could then perhaps utilize the variation between EDs to establish achievable benchmarks of prescribing for the "possibly appropriate" category (e.g., the 20th percentile of prescribing). 

3) Given that the authors presumably have data on prescription duration and whether the agent was broad spectrum vs not, they could go even further and classify not just whether the indication was antibiotic-appropriate or not, but also whether the duration and choice of antibiotic were appropriate. They could calculate the proportion of antibiotics prescribed that had any deficit (inappropriate indication, inappropriate duration, inappropriate agent). These are questions that are really difficult to answer using the administrative datasets that most studies of antibiotic prescribing rely on.

My point is simply that demonstrating variation alone does not move the needle very much, because there will always be variation. It is simply a shame in my mind to use the amazing data the authors have collected simply to document the existence of variation.

Other issues

1) Why focus only on children with fever or a history of fever? Many antibiotics are inappropriately written to children to afebrile children with, say, a 10-day course of non-improving cough.

2) The authors say that they ran a multilevel logistic regression model with clustering on the hospitals. Those two terms don't go together - one typically can either regarding clustering as a nuisance and adjust for it (e.g., generalized estimating equations) vs view the variation between clusters as being of primary interest (multilevel model). I'm pretty sure what the authors did is run a multilevel logistic model with a random hospital intercept (different from clustering on hospital)", but this should be clarified.

3) I can't tell if this is in the model or not, but one of the patient-level predictors should be current fever in the ED vs no fever but positive history - clinicians are definitely more likely to prescribe antibiotics if the child has a fever in the ED. On that note, please write out the regression model - it is not clear from the text what variables are included or not.

4) I think it would be more sensible to restrict the sample only to treat-and-release ED patients - examining antibiotics started in the ED for hospitalized patients seems a bit odd.

Reviewer #2: Thank you for the opportunity to read and review with interest the results of the PERFORM study, a multi-centre observational study describing antibiotic prescribing rates for children < 18 years across several large teaching hospitals in Europe. The study is largely descriptive in nature, quantifying variation in prescribing across hospital centres, and describing the nature of the prescribing behaviour (by type) and consideration of patient and hospital factors which be associated with prescribing rates. The results are topical as growing antibiotic resistance is a continuing problem in many health care systems and reports such as these are very helpful is quantifying the scale of the issue. I have a few comments in regards to mostly the methodology and some suggestions which the authors may want to consider:

To help improve clarity in abstract reporting:

1) Methods - when describing primary outcomes, it is unclear what the denominator is in the rates and what type of unit of measurement this is: i.e. general prevalence as a %, per specific number. Please include how prescription rate is defined in the abstract.

2) Findings - "after standardisation" - It would be helpful to describe how rates were standardisation as there are several different methods of standardisation rates (i.e. direct/indirect)

3) Findings Line 92 - The abstracts reads as if broad and narrow spectrum antibiotics were supposed to be compared but the authors seem to report range combined. Please clarify

4) General comment on abstract findings reporting - Could the authors comment what constitutes descriptors such as "considerable variation", "varied substantially", and "little variation" as there is no a priori definition of this or explanation. I would generally favour refraining from using these adjectives without defining the range differences. For instance, in the results sections, the focus should be on just simply presenting the results (i.e "the range was: "). Instead, the qualitative interpretation of what constitutes "considerable", "little", variation should be provided in the context of the discussion. Alternatively, the authors wish to do this, they should define range definitions in the methods section. 

In the main text:

5) Lines 151-152: This is quite important process of the study design - that was active data collection during the study period with training of research staff on data collection and clinical assessment. Hence, you could even argue there is an interventional component to this analyses and not purely an "observational" study. Whilst I recognise the rationale for this is to improve data capture and quality - the authors primary limitation is further amplified - that these hospitals may not necessary generalise because of this reason. What would have enhanced this would have been to have a pure retrospective one-year look-back period prior to the study active data collection taken place. Whilst it may not possible to do this, it is worth mentioning commenting in the discussion. 

6) Lines 154: I'm am quite puzzled by the sample size justification. The rule-of-thumb of 10 events per variable sample size calculation generally is used as a justification for developing prognostic models. Besides the 10 EPV rule-of-thumb being outdated for prognostic models (see Riley et al. - https://onlinelibrary.wiley.com/doi/full/10.1002/sim.7992), the author applied a multi-level model which means a design effect needs to accounted for (i.e. ICC between hospitals). Further, this study is also not testing a particular hypothesis as purely observational (primary outcome is simply a prescription rate) - so why not just determine the sample size based on the expected/desired precision on the prescribing rates?

7) Lines 224: How many children were excluded with missing data on antibiotic use, presumed cause of infection, and focus of infection and did their characteristics differ from those who were included. It would be good to show if these children were similar or different to the analysis cohort

8) Lines 225-226: What was the rationale for including only the first visit for patients within the first five days? Is there any risk of under-ascertainment of prescribing rates if the initial visit did not include any prescribing but it subsequently occurred at subsequent visits?

9) Lines 229: Use of multi-level logistic regression model needs some justification. For prescribing rates, you could also argue that a Poisson regression model and then incorporate patient level covariates in the model. This would essentially derive your rate over the period of time of data collection as well. 

10) Lines 230-231: How were patient level factors selected for inclusion in the model. By a priori specification or was there any covariate testing for interaction and confounding. Please describe.

11) Lines 250-251: The standardisation process, what was the reference population used to obtain the expected antibiotic prescribing and where was this data obtained from?

12) Lines 260-265: How did the authors define categories of "high" and "low" for their heat maps. What this based on evidence or consensus (i.e. planned analysis, post-hoc consensus). If would be useful to see the pre-specified analytical plan (if there was any) and if there wasn't, this also needs to be mentioned and commented in the discussion as a potential limitation. 

13) Table 1, Final diagnosis is interesting as most look majority are labelled probable viral (57.2%), where definite bacterial only (4.1%) and probable bacterial is 18.1%. It would be useful if the authors could provide a metric of inappropriate or appropriate prescribing. This would require essentially mapping the prescription to the final diagnoses. This analysis stratified by hospital would be useful as Figure 3 shows the three UK hospital having the hospitals having the high rates of prescribing, but at the moment it's not conclusive whether high prescribing was due to having more patients with particular types of diagnoses. 

14) Lines 325: Could this be due to the appropriateness of the prescribing. Can the authors comment on this?

Reviewer #3: This is a large prospective, multicenter observational study that assesses antibiotic prescription in children in 12 European EDs. Data collection was mostly based on routine clinical data, which makes this study particularly interesting. Since adequate disease classification for pediatric infectious diseases is not available for pediatric patients, the authors undertook the effort of classifying the likely etiology of infection. This study in an important first step towards more systematic data collection on antibiotic prescription in European ambulatory settings and provides important data on variability of antibiotic prescription practices. The authors should be congratulated on this important effort. The present manuscript focuses on patient-related factors to describe variations in the prescription rates observed. The manuscript would be greatly strengthened by taking more provider, health systems and data quality issues into account (or to at least give the reader a sense to which extent these were taken into consideration). Specifically, the following points should be addressed further: 

* This study relied on routine data collection. EDs are busy places with high staff turnover. I imagine that the EMR system varies a lot from hospital to hospital, as does the type and quality of routine medical documentation. One would like a better sense on how data was extracted from routine data sources and how the quality of this data was assessed? What are the influencing factors for data quality and completeness? The authors also mention that enumerators were trained in a CRF. I assume that not all routine providers were trained in data collection procedures for this study? How was the data extracted from the routine medical record system into the eCRF? How are antibiotic prescriptions recorded at the different sites? Are there differences in the completeness/ type of documentation between ambulatory and admitted patients? In some countries, for example, DRGs are implemented for admitted patients resulting in differential documentation for hospitalized patients due to billing issues. 

* What are provider-related factors that influence prescription between sites (type of provider, level of training)? Please represent in your data analysis.

* I imagine that antibiotic prescription is largely related to local/national guidelines. For example, nitrofurantoine is specifically not recommended as first line for UTIs in some countries to spare the antibiotic; and TMP/SULFA is recommended as a first line instead. How was this assessed in the study? Could you add an analysis: per guideline/ not per guideline?

* What are health-system related factors beyond guidelines that drive antibiotic prescription? Which ones were assessed/ not assessed?

* I understand how the classification of antibiotics into narrow versus broad was reached; however, it is certainly disputable in some instances (e.g. pipercillin =narrow, cefuroxim=broad). Could you provide some 

Furthermore, a better sense of the generalizability of the data should be provided. How was the site selection performed? How does it relate to national averages in key institutional/ health system aspects?

Are you looking at use or prescriptions? Two different concepts that are measured differently. Please clarify across the manuscript. 

Additional Minor Comments

* L113: the IQR appears out of context here. Specify

* Introduction: give short introduction to PERFORM project

* AIM: reformulate aim to align with your analysis. I think you would want to say: variations in rate and types of antibiotic prescription

Methods

* How were the participating hospitals selected?

* Are these pediatric EDs? Who sees patients? GPs? Pediatricians? Emergency Medicine Specialists?

* How was duration of data collection determined?

* The rationale for your sample size calculation is unclear to me; please expand. Comment on generalizability

* Does the missing at random assumption really hold here (for example various NICE traffic light factors are related amongst each other)? How did you decide on this?

* Did you exclude patients with missing data on antibiotic use or antibiotic prescription?

* How did you account for the difference in triage systems to account for "triage level"?

* "Second, we used multilevel logistic regression using clustering on hospital to study variation of antibiotic use between hospitals". This is unclear to me: if you introduce hospital as a random effect (clustering), you would not be able to compare between hospitals?

* How did you decide on the CRP cutoffs? Based on what?

Results

* Table 1: the range of patients classified under the triage category seems very wide. How is this explained?

* L 359: remained…. Is there a word missing?

* Fog 5/6: heat map of what?

[LINK]

---

## [Decision Letter · Decision Letter 1]

31 May 2020

Dear Dr. Moll,

Thank you very much for re-submitting your manuscript "Variation in antibiotic prescription rates in febrile children presenting to Emergency Departments across Europe (PERFORM): a multicentre observational study" (PMEDICINE-D-19-04008R1) for review by PLOS Medicine.

I have discussed the paper with my colleagues and the academic editor and it was also seen again by two reviewers. I am pleased to say that provided the remaining editorial and production issues are dealt with we are planning to accept the paper for publication in the journal.

[LINK]

We look forward to receiving the revised manuscript by Jun 05 2020 11:59PM. 

Sincerely,

Thomas McBride, PhD

Senior Editor 

PLOS Medicine

plosmedicine.org

Requests from Editors:

1- Thank you for agreeing to make your data available in a public data repository. At this time, please provide the url of the data repository, the DOI for the dataset, and any other information needed for readers to access the dataset (including the data contact email already provided).

2- Thank you for editing your title. Is it more appropriate to include MOFICHE than PERFORM, though?

3- The url for PERFORM can be removed from the Abstract and supplied in the Introduction or Methods.

4- In the Abstract Methods and Findings section, please include the overall antibiotic prescription rate (all hospitals combined) along with the ranges. Please also specify that the ranges are across the different hospitals.

5- Thank you for providing more details in the Abstract Conclusions. This section could be a bit more succinct. For example, the second and third sentences are redundant given the previous section.

6- Please edit the last sentence of the Abstract Conclusions to read: “Until better diagnostics are available to accurately differentiate between bacterial and viral aetiologies, implementation of antimicrobial stewardship guidelines across Europe is necessary to limit antimicrobial resistance.”

7- Thank you for adding an Author Summary. The second point can be removed.

8- Additionally, please edit the “What Did the Researchers Do and Find?” section of the Author Summary could be edited for brevity, perhaps focusing on the standardized prescription rates and the prescriptions with inappropriate indication or duration.

9- Please also add “In this study we found…”, “These findings suggest...”, or similar to the “What Do These Findings Mean?” section of the Author Summary.

10- Thank you for noting ethics approval. Please provide a list (Supplemental Text is fine) of the ethics committees from the participating hospitals.

11- Please make reference to the S4 appendix (details of the regression model) in the main text Methods section.

12- S7 Appendix: typo in the title.

13- S12, S14 Appendix: Please make sure the title in the file matches the title in the list of supplemental files in the main text.

14- S15 Appendix is a file with tracked changes, please supply a clean version.

15- The data presented in Figure 2 would be better presented as a table. Please revise accordingly. 

16- Please revise the stacked bar chart shown in Figure 3 as a side-by-side bar graph.

17- Figure 6: include Otitis media and Tonsillitis/pharyngitis in the title?

18- Figures 4-7: it seems there is plenty of room to provide 95% CIs in these fgures without asking the reader to open a supplementary file. Please consider adding.

19- Figures 4-8: the legends are not much use without a scale for the gradient.

20- Discussion, second paragraph: “Our study supports previous studies that reported variable antibiotic prescribing…”?

21- Discussion, final paragraph: “Until better diagnostics are available to accurately differentiate between bacterial and viral aetiologies, we strongly urge the implementation of antimicrobial stewardship guidelines to reduce antibiotic prescription in febrile children across Europe.”

22- Please move the mention of the analysis plan and the STROBE checklist to earlier in the Methods so that they appear as S1 and S2.

Comments from Reviewers:

Reviewer #1: 1) I appreciate the efforts that the authors have made to respond to the comments. I remain concerned, however, about the unclear implications of all the variation they document. The authors write in the introduction that "understanding variability is essential for the development and implementation of interventions to optimize antibiotic use." Why? They never explicitly state what the implications would be if they showed low variability versus high variability on each of their outcomes.

It appears that the five main outcomes are as follows: 1) the antibiotic prescribing rate; 2) the proportion of antibiotics that were broad-spectrum; 3) the % of antibiotics that were for appropriate (presumed bacterial), inappropriate (presumed viral) or of unclear appropriateness (viral or bacterial); 4) the % of antibiotics that were of inappropriate duration; and 5) the % of antibiotics for urinary tract infections and respiratory infections that were guideline-concordant. Here are my view of the implications of variation in each of these outcomes.

- Variability in overall antibiotic prescribing rate. Documenting variability in this rate is not that helpful in my view. The authors seem to implicitly believe that the variation implies overprescribing, but since the ideal rate of prescribing is unknown, it is unclear if the high prescribing EDs are overprescribing, if the low prescribing EDs is underprescribing, or both. 

- Variability in broad-spectrum prescribing - same as above. The ideal rate of broad-spectrum prescribing is unknown. 

- Variability in prescribing for antibiotic-inappropriate conditions (e.g., viruses) - documenting variability is helpful because the ideal rate of use is known (0%). If there is a lot of variation with some very high prescribing EDs, it would imply that some EDs that are doing way worse than others and therefore quality improvement initiatives should perhaps take a more targeted approach focused on the outliers. If there is little variation and the rate of prescribing among EDs is high, that would imply a more global approach is needed. If there is little variation and the rate of prescribing among EDs is consistently closer to 0%, then maybe we don't need to worry about overprescribing for viral infections.

Note that I don't feel the same about prescribing for bacterial infections - how would finding variation inform stewardship initiatives? I'm not sure it would, so it's not clear this analysis needs to be included. Also, variability in prescribing for unknown bacterial/viral infection is hard to interpret because the ideal rate of prescribing is unknown, so I'm also not sure it needs to be included.

- Variability in prescribing of inappropriate duration - this is helpful because we know the rate should be 0%

- Variability in guideline concordance - this is helpful because we know the rate should be 100%. 

I would urge the authors to more explicitly walk the reader through the implications of variation in each outcome in the Discussion, using the above as points to consider incorporating. Also, in the introduction, it would be helpful to motivate the analysis by arguing that assessing the degree of variation on outcomes that should be 0% or 100% can inform whether quality improvement initiatives should be global versus targeted.

2) Presentation issues

- Abstract is hard to follow. For example, it reports guideline-concordant prescribing but does not mention earlier that this will be an outcome. It comes out of nowhere. 

I would restructure as follows. State that the main outcomes are 1) the antibiotic prescribing rate; 2) the proportion of antibiotics that were broad-spectrum; 3) the % of antibiotics that were for appropriate (presumed bacterial), inappropriate (presumed viral) or of unclear appropriateness (viral or bacterial); 4) the % of antibiotics that were of inappropriate duration; and 5) the % of antibiotics for urinary tract infections and respiratory infections that were guideline-concordant. Then state that the overall rate across all children was calculated. Then state that for each outcome, variation was assessed by calculating standardized prescribing rates (i.e., the ratio between observed and expected antibiotic prescribing rate) using multilevel logistic regression models. Then in the results, report the overall result and the range in standardized prescription rates for each of the five outcomes.

- Abstract and discussion both state that antibiotic stewardship initiatives are needed until better diagnostics are available. That's simply not true. Even with perfect diagnostics, we would still need stewardship initiatives (e.g., to ensure appropriate selection and duration of antibiotics).

- In general, the manuscript is also difficult to follow. There are a lot of analyses, and they are not always "signposted" well. It would be helpful to have a section in the Methods called "Outcomes" that defines each of the five main outcomes. Then have a section called "Statistical analysis" that describes the statistics used to assess the outcomes overall and to describe variation. 

- I don't think the antimicrobial resistance analysis is that helpful - it's distracting and comes out of nowhere in my view. Consider deleting.

- The prescribing for respiratory tract infections also seems a little bit extraneous, in large part because it's not clear what the ideal rate of prescribing should be for something like ear infections. Consider deleting. Readers will have an easier time digesting this very dense manuscript if there are just the five analyses for the five outcomes and if the analyses are reported in a consistent manner.

Reviewer #2: The authors have conducted a very thorough job providing additional analyses and responding to my review comments. The manuscript, upon reading, is much clearer and strengthened. I have no additional points to raise.

[LINK]

---

## [Editor Report · Decision Letter 2]

28 Jul 2020

Dear Prof. Moll, 

On behalf of my colleagues and the academic editor, Dr. Jean-Louis Vincent, I am delighted to inform you that your manuscript entitled "Variation in antibiotic prescription rates in febrile children presenting to Emergency Departments across Europe (MOFICHE): a multicentre observational study" (PMEDICINE-D-19-04008R2) has been accepted for publication in PLOS Medicine. 

PRODUCTION PROCESS

PRESS

PROFILE INFORMATION

Thank you again for submitting the manuscript to PLOS Medicine. We look forward to publishing it. 

Best wishes, 

Thomas McBride, PhD

Senior Editor 

PLOS Medicine

plosmedicine.org